# Single-atom catalysts reveal the dinuclear characteristic of active sites in NO selective reduction with NH$_3$

Weiye Qu [1,3], Xiaona Liu[1,3], Junxiao Chen[1], Yangyang Dong[1], Xingfu Tang[1,2✉] & Yaxin Chen [1✉]

High-performance catalysts are extremely required for controlling NO emission via selective catalytic reduction (SCR), and to acquire a common structural feature of catalytic sites is one key prerequisite for developing such catalysts. We design a single-atom catalyst system and achieve a generic characteristic of highly active SCR catalytic sites. A single-atom Mo$_1$/Fe$_2$O$_3$ catalyst is developed by anchoring single acidic Mo ions on (001) surfaces of reducible α-Fe$_2$O$_3$, and the individual Mo ion and one neighboring Fe ion are thus constructed as one dinuclear site. As the number of the dinuclear sites increases, SCR rates increase linearly but the apparent activation energy remains almost unchanged, evidencing the identity of the dinuclear active sites. We further design W$_1$/Fe$_2$O$_3$ and Fe$_1$/WO$_3$ and find that tuning acid or/and redox properties of dinuclear sites can alter SCR rates. Therefore, this work provides a design strategy for developing improved SCR catalysts via optimizing acid-redox properties of dinuclear sites.

[1] Department of Environmental Science & Engineering, Fudan University, 200433 Shanghai, China. [2] Jiangsu Collaborative Innovation Center of Atmospheric Environment & Equipment Technology, Nanjing University of Information Science & Technology, 210044 Nanjing, China. [3] These authors contributed equally: Weiye Qu, Xiaona Liu. ✉email: tangxf@fudan.edu.cn; 16110740025@fudan.edu.cn

Selective catalytic reduction (SCR) of NO with NH$_3$ over V$_2$O$_5$-based catalysts is a widely used technology for controlling NO emission from stationary sources[1]. The increasingly stringent emission regulations demand the development of high-performance SCR catalysts available for various harsh conditions. However, a common structural feature of active catalytic sites (ACSs), as well as reaction mechanisms, is still obscure[2–6], which becomes one of the main obstacles for developing such catalysts.

The nature of ACSs has been extensively studied to achieve the generic feature of ACSs to develop highly active catalysts since V$_2$O$_5$-based catalysts were applied for SCR in 1970s. It is commonly accepted that highly active catalysts require ACSs to simultaneously possess acid-redox features[2–4], but the origins of the acid-redox properties, provided solely by one mononuclear site or respectively by two adjacent metal sites, i.e., one dinuclear site, are still highly debated[2,3,7,8]. Even for typical V$_2$O$_5$-based catalysts, there is little consensus on the structure of ACSs. Marberger et al.[3] identified ACS as one mononuclear vanadium site, which serves not only as one Lewis acid site for NH$_3$ adsorption, but also as one redox site to close a SCR cycle, similar to the ACS structure proposed by Ramis et al.[7]. However, Topsøe et al.[2,9,10]. identified an adjacent dinuclear vanadium site as ACS, which was subsequently used as a structural model to describe SCR mechanisms[1,5]. Went et al.[8,11] confirmed the coexistence of monomeric vanadyl and polymeric vanadate species including dimers on V$_2$O$_5$/TiO$_2$ surfaces, all of which were catalytically active in SCR. These discrepancies in identification of ACSs have mainly arisen from site-averaged information obtained from the studied catalysts without uniform active sites.

Single-atom catalysts with uniform ACSs are favorable for studying the nature of ACSs[12–14]. Wark et al.[13] reported that a single-atom V$_1$/ZSM-5 catalyst had significant SCR activity, implying the mononuclear active sites, but its activity is much lower than that over V$_2$O$_5$/TiO$_2$ with abundant dinuclear sites, in line with the results that dinuclear sites are superior to mononuclear sites[8,15,16]. Although the existence of the dinuclear sites above is either experimentally speculative or theoretically predicted, dinuclear ACSs appear to be necessary for high SCR activity. Likewise, a dynamically formed transient Cu dimer showed higher SCR rates than one Cu monomer[14], indicating the requirement of dinuclear ACSs in SCR. An ideal strategy is to fabricate dinuclear metal sites on supports[17], which allows them to function as dual sites catalyzing SCR reaction, but it is a formidable task to synthesize such a catalyst.

Here, we develop a single-atom Mo$_1$/Fe$_2$O$_3$ catalyst, and thus the isolated acidic Mo ions and one adjacent surface redox Fe ions are assembled as uniform dinuclear acid-redox sites, which shows high SCR turnover frequencies (TOFs) comparable to V$_2$O$_5$/TiO$_2$. To tune acid-redox properties of dinuclear sites, we develop W$_1$/Fe$_2$O$_3$ and Fe$_1$/WO$_3$, and find that SCR activity can be controlled by tuning acid or/and redox properties of dinuclear sites, thus implying a common dinuclear feature of highly active catalytic sites.

## Results

### Fabrication of the single-atom Mo$_1$/Fe$_2$O$_3$ catalyst.
We prepared hexagon-shaped α-Fe$_2$O$_3$ nanosheets mainly exposing {001} facets[18], as confirmed by synchrotron X-ray diffraction (SXRD, Supplementary Fig. 1) and transmission electron microscopy (TEM, Supplementary Fig. 2) techniques. On the Fe$_2$O$_3$(001) surface, there are plenty of threefold hollow sites formed by three surface lattice oxygen atoms (Supplementary Fig. 2), which serve as suitable sites for anchoring Mo$^{5+/6+}$ or W$^{5+/6+}$ with an ionic radius of ~0.6 Å[19]. We successfully anchored single Mo ions on

the Fe$_2$O$_3$(001) surfaces to get a single-atom Mo$_1$/Fe$_2$O$_3$ catalyst (Fig. 1). The highly dispersed Mo ions are evidenced by the energy dispersive X-ray spectroscopy (EDX) mappings of Mo$_1$/Fe$_2$O$_3$ (Fig. 1c–f) and the SXRD patterns of Mo$_1$/Fe$_2$O$_3$ (Supplementary Fig. 1). In Fig. 1g, the aberration-corrected high-angle annular dark-field scanning TEM (AC-STEM) image of Mo$_1$/Fe$_2$O$_3$ shows that the Mo ions are atomically dispersed on the α-Fe$_2$O$_3$(001) surface. As further analyzed by the selected-area intensity surface plot and the corresponding structural model (Fig. 1h and Supplementary Fig. 3), the Mo ions are precisely anchored on the threefold hollow sites (the yellow circles in Fig. 1g, h). Hence, each isolated Mo ion and one adjacent outermost surface Fe ion (denoted as Fe$_{surf}$ in Fig. 1h) with a distance of ~2.9 Å in between are assembled as one dinuclear site (the red ellipse in Fig. 1g).

### Structures of the dinuclear site.
Figure 2a shows the χ(R) $k^2$-weighted Fourier-transform extended X-ray absorption fine structure (FT-EXAFS) spectra of Mo$_1$/Fe$_2$O$_3$ and α-MoO$_3$ at the Mo K-edge and α-Fe$_2$O$_3$ at the Fe K-edge (Supplementary Figs. 4 and 5), and the related structure parameters are listed in Supplementary Table 1. The FT-EXAFS spectrum of Mo$_1$/Fe$_2$O$_3$ is similar to that of α-Fe$_2$O$_3$, implying that the Mo ion is located at a surface site corresponding to the Fe site in α-Fe$_2$O$_3$ bulk, but distinctly different from that of α-MoO$_3$, ruling out the existence of α-MoO$_3$ on Mo$_1$/Fe$_2$O$_3$ surfaces. The second peak is contributed from the scattering path between the Mo atom and the neighboring Fe atoms. An average distance between Mo and Fe is ~2.93 Å with a coordination number (CN) of 3 (Supplementary Table 1)[20], consistent with the observation of the dinuclear Mo$_1$-Fe$_1$ site in Fig. 1g. The first peak can be assigned to the Mo–O bonds with an average bond length of ~1.88 Å and a CN of 6 (Supplementary Table 1), indicating the existence of a MoO$_6$ motif, i.e., each anchored Mo ion has three surface dangling bonds besides three Mo–O bonds formed with three oxygen ions of the anchoring site. The Mo–O bond length (1.88 Å) is shorter than the Fe-O bonds (1.98 Å) in α-Fe$_2$O$_3$ (Supplementary Table 1), which shows the existence of surface dangling Mo = O bond(s), as evidenced by a double-bond-specific Raman band at ~989 cm$^{-1}$ appearing in the Raman spectrum of Mo$_1$/Fe$_2$O$_3$[21] (Supplementary Fig. 6).

In the Mo $L_3$-edge X-ray absorption spectra of α-MoO$_3$ and Fe$_2$(MoO$_4$)$_3$ (Fig. 2b), two peaks of α-MoO$_3$ are readily assigned to the Mo $2p_{3/2} \rightarrow 4d(t_{2g}^0)$ and $4d(e_g^0)$ transitions, respectively, with a ligand-field splitting energy of ~3.1 eV for a MoO$_6$ octahedral symmetry ($O_h$)[21], which reduces down to ~1.9 eV for a MoO$_4$ tetrahedral symmetry ($T_d$) in Fe$_2$(MoO$_4$)$_3$[21]. The Mo $L_3$-edge X-ray absorption spectrum of Mo$_1$/Fe$_2$O$_3$ is characteristic of two peaks with a splitting energy of ~3.1 eV (Fig. 2b), similar to that of α-MoO$_3$. These results combined with the above AC-STEM image, the EXAFS data and the Raman evidence manifest the existence of a distorted MoO$_6$ octahedral structure on the Mo$_1$/Fe$_2$O$_3$ surface.

### Acid and redox properties of the dinuclear site.
The absorption peak of Mo$_1$/Fe$_2$O$_3$ due to the Mo $2p_{3/2} \rightarrow t_{2g}$ transition is weaker than that of α-MoO$_3$ when we normalized the intensity of their peaks due to the Mo $2p_{3/2} \rightarrow e_g^0$ transitions (Fig. 2b), implying that the oxidation state of the isolated Mo ion is lower than Mo$^{6+}$ in α-MoO$_3$. More accurately, owing to the peak areas proportional to the unoccupied states of the orbitals[22], we deconvoluted the spectrum of Mo$_1$/Fe$_2$O$_3$ to two individual peaks, and thus the unoccupied states of the $t_{2g}$ and $e_g$ orbitals are positively proportional to the areas of the blue shade and the red shade (Fig. 2b), respectively. An area ratio of the two peaks is ~5:4, indicating an electronic configuration ($t_{2g}^1 e_g^0$) of the Mo $4d$

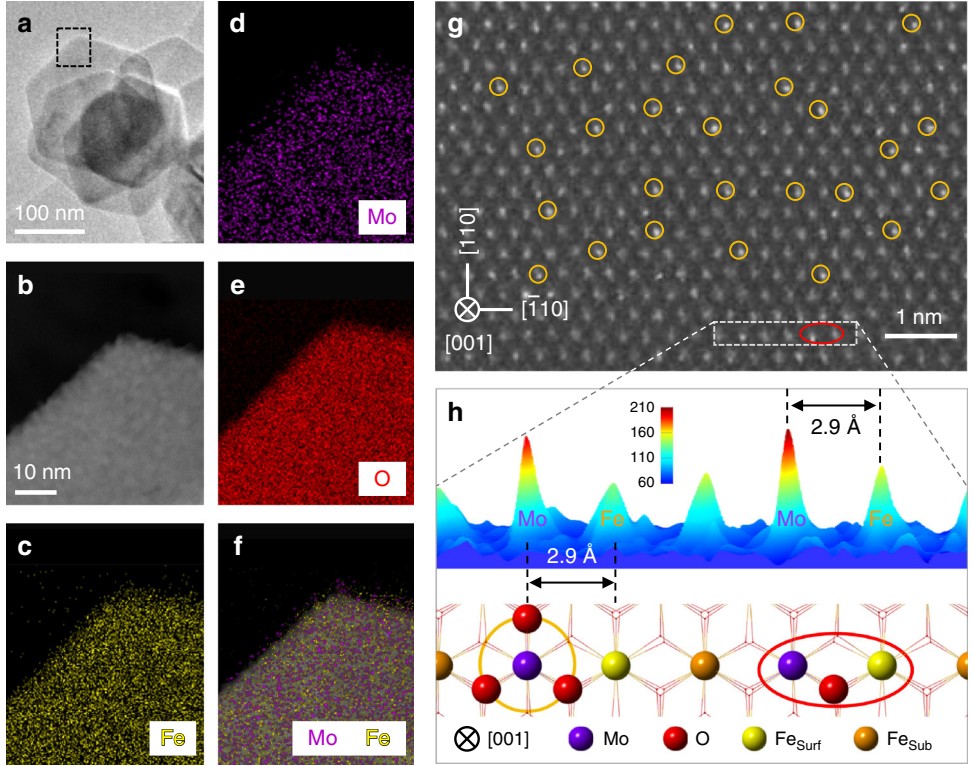

**Fig. 1 EDX mapping and AC-STEM images of Mo₁/Fe₂O₃.** **a** TEM image of $Mo_1/Fe_2O_3$. **b** AC-STEM, and **c–f** EDX mapping images of the selected area (black dashed rectangle) in **a**. **g** AC-STEM image of $Mo_1/Fe_2O_3$. **h** Intensity surface plot and the corresponding structural model of the selected area (white dashed rectangle) in **g**. The Mo loading is 1.3 wt% with respect to $\alpha$-$Fe_2O_3$. The purple, red, yellow, and brown balls represent Mo atoms, O atoms, surface Fe atoms ($Fe_{surf}$), and subsurface Fe atoms ($Fe_{sub}$), respectively. Selected single Mo atoms and dinuclear $Mo_1$-$Fe_1$ sites are marked by the yellow circles and the red ellipses, respectively.

orbitals of the Mo ions of $Mo_1/Fe_2O_3$, i.e., the Mo species are $Mo^{5+}$, in accordance with the data of the Mo $3d$ X-ray photoelectron spectrum of $Mo_1/Fe_2O_3$ (Supplementary Fig. 7). For the $MoO_6$ motif, three oxygen atoms are provided by electroneutral $\alpha$-$Fe_2O_3$, and the remaining fragment is negatively charged $[MoO_3]^-$ due to $Mo^{5+}$ and $O^{2-}$. Taking the nearly unchanged oxidation state of Fe after the Mo anchoring into account (Supplementary Fig. 8), the $MoO_6$ motif contains one hydrogen ion ($H^+$) for the charge balance, as evidenced by the diffuse reflectance infrared Fourier-transform (DRIFT) spectra in Supplementary Fig. 9. Thus, each isolated Mo ion due to the formation of a $MoO_6H$ species can provide one Brønsted acid site[23], which can transform to the Lewis acid site during the SCR reactions[24] or in the SCR temperature window (Supplementary Fig. 10).

Apart from the acidic property, the redox ability of ACS is also required for closing a SCR cycle[2]. To study the redox property of $Mo_1/Fe_2O_3$, we carried out $H_2$ temperature-programmed reduction procedure ($H_2$-TPR) for three samples (Fig. 3a). $\alpha$-$Fe_2O_3$ has a much stronger reduction ability than $\alpha$-$MoO_3$. A shoulder peak of $\alpha$-$Fe_2O_3$ at ~310 °C is readily attributed to the $Fe_2O_3 \rightarrow Fe_3O_4$ reduction, and a strong peak at ~380 °C with a discernible shoulder at a high-temperature edge (~410 °C) can be due to the $Fe_3O_4 \rightarrow Fe^0$ reduction[25]. Subtly, a very weak shoulder appears in a temperature regime 180–320 °C (up-left inset of Fig. 3a), which can be assigned to the reduction of the surface active oxygen of $\alpha$-$Fe_2O_3$ (Supplementary Discussion). The Mo anchoring has little effect on the redox ability of the surface active O atoms of $\alpha$-$Fe_2O_3$ (up-left inset of Fig. 3a), but the reduction temperatures of the bulk O atoms shifted up by ~20 °C. Moreover, an extra weak peak appears in a high-temperature regime 460–615 °C

(up-right inset in Fig. 3a), which can be readily assigned to the reduction of the Mo species (Supplementary Discussion). This result evidences that the redox property of $Mo_1/Fe_2O_3$ originates from the $FeO_x$ species of the dinuclear sites. Since the Mo and Fe ions can provide the acidic and redox properties, respectively, the dinuclear site possesses the common acid-redox properties of ACS of SCR[2].

**Identifying the dinuclear characteristic of active sites.** To identify active sites of $Mo_1/Fe_2O_3$, we synthesized a series of $Mo_1/Fe_2O_3$ with the different number of the Mo ions by tuning the Mo loadings (Supplementary Discussion). NO conversions ($X_{NO}$) in SCR over these $Mo_1/Fe_2O_3$ catalysts are shown in Supplementary Fig. 11, and $X_{NO}$ increases with the Mo loading, whereas the reducible $\alpha$-$Fe_2O_3$ and the acidic $\alpha$-$MoO_3$ alone have very low SCR activities under identical reaction conditions. Meanwhile, $N_2$ selectivity of $\alpha$-$Fe_2O_3$ drastically enhances after the Mo loading (Supplementary Fig. 12), and the $Mo_1/Fe_2O_3$ also shows excellent $H_2O$ and/or $SO_2$ durability (Supplementary Fig. 13). This evidences that both the acid site and the redox site are required for excellent SCR performance, consistent with the literature[2–4]. In the reaction kinetics regime ($X_{NO} < 15\%$), we extracted SCR rates at 270 °C from Supplementary Fig. 11, and the apparent activation energy ($E_a$) from the Arrhenius plot of Supplementary Fig. 14. In Fig. 3b, SCR rates increase linearly with the Mo number, and $E_a$ almost remains constant ($86 \pm 4$ kJ mol$^{-1}$), evidencing that the active sites of $Mo_1/Fe_2O_3$ are uniform. Owing to the acid-redox properties and the similar structure as the dinuclear active site of $V_2O_5$-based catalysts[2,26], and the same catalytic behavior as the dinuclear Cu sites[14], the dinuclear $Mo_1$-$Fe_1$ sites are rationally

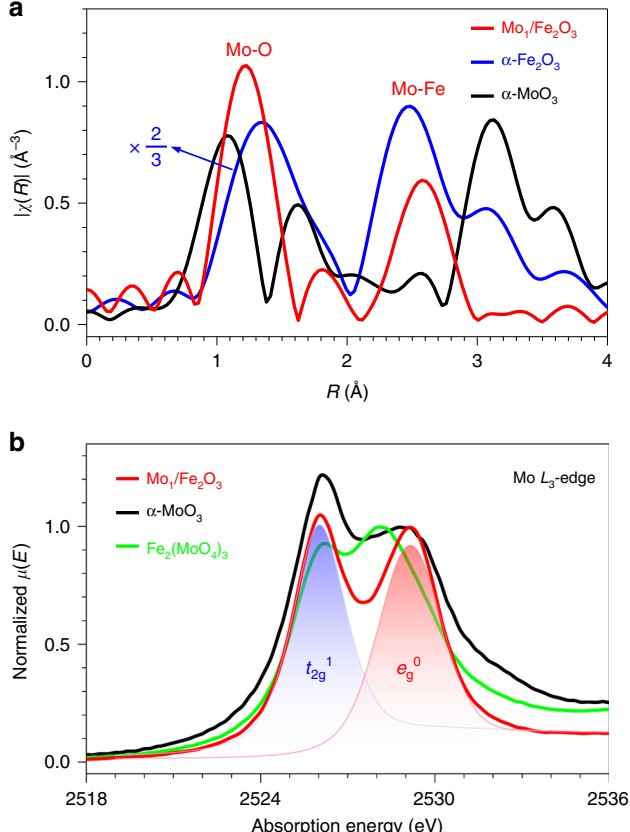

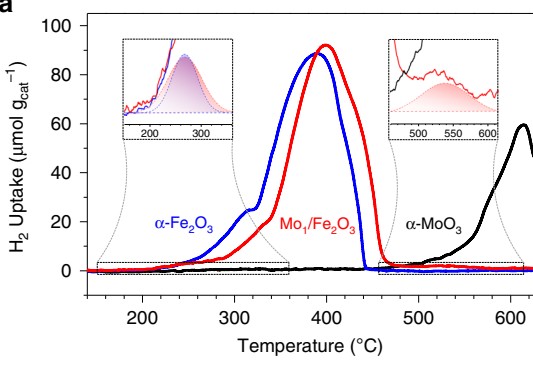

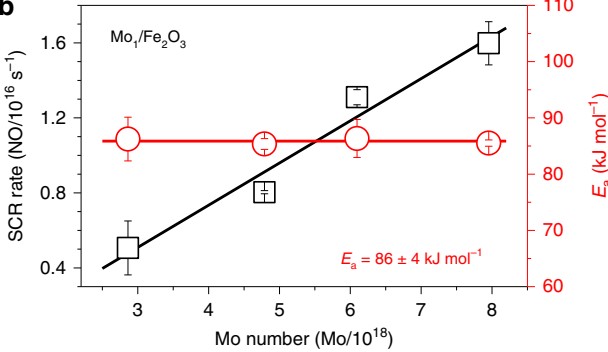

**Fig. 2 The geometric structure of the MoO$_6$ motif and the electronic structure of the isolated Mo ions. a** $\chi(R)$ $k^2$-weighted FT-EXAFS spectra of Mo$_1$/Fe$_2$O$_3$ (red line) and α-MoO$_3$ (black line) at the Mo $K$-edge together with α-Fe$_2$O$_3$ (blue line) at the Fe $K$-edge. **b** Mo $L_3$-edge X-ray absorption spectra of Mo$_1$/Fe$_2$O$_3$ (red line), α-MoO$_3$ (black line) and Fe$_2$(MoO$_4$)$_3$ (green line). Blue and red shades represent the unoccupied states of the Mo $t_{2g}$ and $e_g$ orbitals, respectively.

**Fig. 3 Redox property of dinuclear sites and identification of the active sites. a** H$_2$-TPR profiles of Mo$_1$/Fe$_2$O$_3$ (red line) with a 1.3 wt% Mo loading, α-Fe$_2$O$_3$ (blue line) and α-MoO$_3$ (black line). Insets: enlargements of the selected regimes together with the fitting data, and the curves of α-MoO$_3$ and α-Fe$_2$O$_3$ are omitted in the up-left and up-right insets, respectively, for clarity. **b** SCR rates (black square) at 270 °C and $E_a$ (red circle) on Mo$_1$/Fe$_2$O$_3$ with the different Mo number. The error bars represent standard error.

identified as ACSs. Otherwise, $X_{NO}$ decreases (Supplementary Fig. 15) when the structure of the dinuclear sites were destroyed (Supplementary Fig. 16). Furthermore, we calculated TOFs (converted NO molecules per ACS per second) to be ~$1.7 \times 10^{-3}$ s$^{-1}$ at 270 °C (Supplementary Fig. 17), comparable to the TOF values ($1.3 \times 10^{-3}$ s$^{-1}$ at 277 °C[27], $2.4 \times 10^{-3}$ s$^{-1}$ at 323 °C[16]) of V$_2$O$_5$/TiO$_2$.

We tuned the acid-redox properties of dinuclear sites, and studied the effect of the properties on SCR activity to substantiate whether the structure feature of dinuclear acid-redox sites can act as a generic structural model of ACSs. To tune the acidic property of the dinuclear site, we anchored the W ions with the weaker acidity than the Mo ions[23] on the α-Fe$_2$O$_3$(001) surfaces to get W$_1$/Fe$_2$O$_3$ (Supplementary Figs. 18 and 19). As shown in the EDX mapping and AC-STEM images of Fig. 4a–d, the isolated W ions are precisely anchored on the threefold hollow sites to construct dinuclear W$_1$-Fe$_1$ active sites, as marked in red ellipses in Fig. 4d. Likewise, the acidic property (Supplementary Fig. 9) and the redox property (Supplementary Fig. 20) of the dinuclear W$_1$-Fe$_1$ active sites originate from the isolated W ions and the Fe ions, respectively. The trend in catalytic activity of W$_1$/Fe$_2$O$_3$ is similar to that of Mo$_1$/Fe$_2$O$_3$ (Supplementary Fig. 21) and $E_a$ remains constant (Supplementary Fig. 22). In Fig. 4e, the linearly increasing SCR rates over W$_1$/Fe$_2$O$_3$ with the number of ACSs show the behavior expected of the dinuclear catalytic sites[14]. Catalytic activities of W$_1$/Fe$_2$O$_3$ are slightly lower than Mo$_1$/Fe$_2$O$_3$

(Supplementary Figs. 23 and 24), which indicates that tuning the acidity of the dinuclear site can alter SCR activities.

To tune the redox property, we anchored single Fe ions on (001) surfaces of the square-shaped γ-WO$_3$ nanosheets[28] to achieve Fe$_1$/WO$_3$ (Supplementary Discussion), the structures of which were characterized by HRTEM (Supplementary Fig. 25). As the atomic number of W ($Z = 74$) significantly overnumbers that of Fe ($Z = 26$) and the very low Fe loadings (Supplementary Discussion), we failed to observe the single Fe atoms on the γ-WO$_3$(001) surfaces by using the AC-STEM imaging and X-ray absorption spectroscopy. The EDX mapping images show the highly dispersed Fe ions on the surfaces (Supplementary Fig. 25). A linear relation between the SCR rate and the number of ACSs (Fig. 4e) and a same trend of catalytic behavior (Supplementary Fig. 26) as that of W$_1$/Fe$_2$O$_3$ or Mo$_1$/Fe$_2$O$_3$ indicate the existence of dinuclear W$_1$-Fe$_1$ sites. The redox property of the dinuclear W$_1$-Fe$_1$ site of Fe$_1$/WO$_3$ is much weaker than that of W$_1$/Fe$_2$O$_3$ (Supplementary Fig. 20), which is one main factor that led to the SCR rates of Fe$_1$/WO$_3$ lower than W$_1$/Fe$_2$O$_3$ (Fig. 4e and Supplementary Figs. 23 and 24). Therefore, to tune the acidic or/ and redox properties of the dinuclear sites can alter SCR activities, indicating that the dinuclear acid-redox site can function as a generic structural model of highly active catalytic sites of SCR.

## Discussion

The dinuclear structural model could provide a basis for a precise identification of highly active SCR catalytic sites. With an assist of this model to identify dinuclear ACSs, it is not difficult to

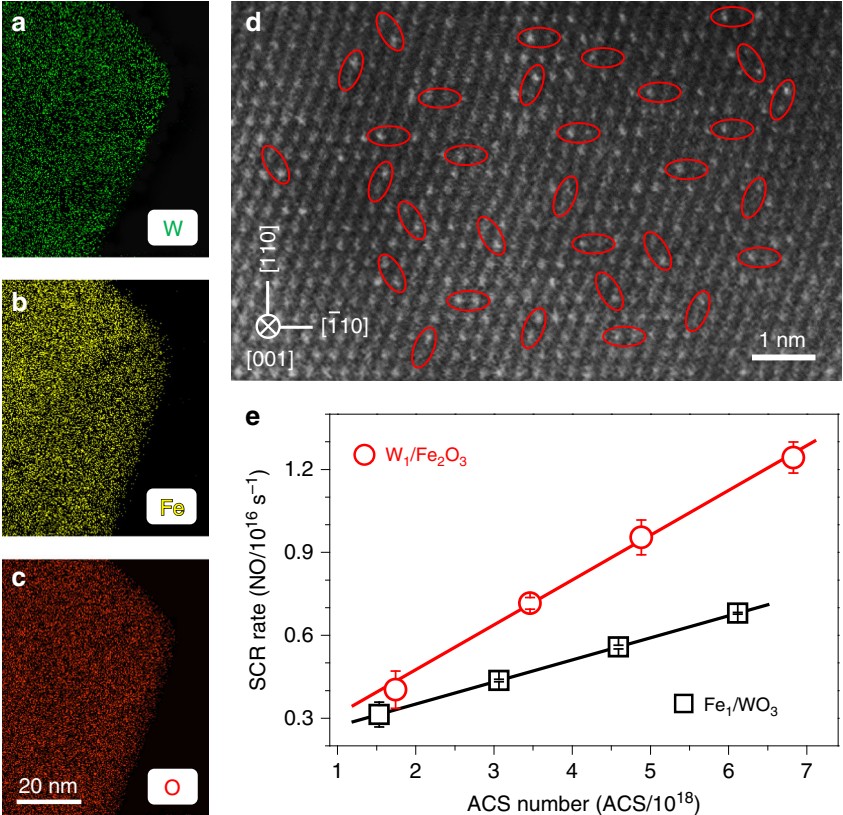

**Fig. 4 The effect of the acid-redox properties of dinuclear sites on SCR activity. a–c** EDX mapping, and **d** AC-STEM images of $W_1/Fe_2O_3$ with a 2.0 wt% W loading. Selected dinuclear $W_1$-$Fe_1$ sites are marked by the red ellipses. **e** SCR rates over $W_1/Fe_2O_3$ (red circle) and $Fe_1/WO_3$ (black square) with different ACS numbers at 270 °C. The error bars represent standard error.

understand the fact that dimeric vanadium sites show much higher SCR rates than monomeric sites for $V_2O_5$-based SCR catalysts[5,8], and that a parabola-type curve of TOFs appears as the vanadium coverage increases[29]. This model also give a satisfactory explanation for the promotional effect of $WO_3$ or $MoO_3$ for $V_2O_5/TiO_2$ mainly because of the emerging dinuclear W-V or Mo-V catalytic sites more active than the dinuclear V-V sites due to the strong acidity properties of W or Mo ions[6,15,23]. This model underpins our fundamental understanding why SCR reactions preferentially occur at interfaces of acid-redox oxide catalysts such as Mo-Fe[30], W-Fe[31], W-Ce[32] oxides and so on, leading to their high SCR rates. We anticipate that after optimizing the acid-redox properties of dinuclear sites, improved SCR catalysts with abundant dinuclear acid-redox sites will be developed for more efficiently controlling NO emission.

## Methods

**Sample synthesis**. All the chemicals are of analytical grade and used as received.

**Single-atom $Mo_1/Fe_2O_3$ catalysts**. We firstly prepared the $\alpha$-$Fe_2O_3$ hexagonal nanosheets according to the reference[18]. $FeCl_3$·$6H_2O$ (0.54 g, 2.0 mmol) was dissolved in ethanol (30.0 mL) with a trace addition of de-ionized water (1.5 mL) under vigorously magnetic stirring until completely dissolved, to which sodium acetate (1.91 g, 23.3 mmol) was added under stirring. The mixture was sealed in a Teflon-lined stainless steel autoclave (50 mL) and maintained in the oven at 180 °C for 24 h. After natural cooling to room temperature, the resulting solid was washed with de-ionized water and ethanol several times, respectively, dried at 60 °C for 4 h. The single-atom $Mo_1/Fe_2O_3$ catalysts were prepared by the impregnation method. The certain amount of $(NH_4)_6Mo_7O_{24}$·$4H_2O$ was solved in de-ionized water to form an aqueous solution, to which the $\alpha$-$Fe_2O_3$ powder (2.00 g) was added under vigorously magnetic stirring at 80 °C until the water was evaporated. Then all the samples were dried at 80 °C for 12 h and calcined at 550 °C in air for 3 h. The Mo amount was adjusted according to the number of the anchoring sites of the $\alpha$-$Fe_2O_3$ surfaces (see the Section 2 of Supplementary Discussion for more details)

to get a series of the samples with the different Mo loadings, and the obtained samples were measured to be 1.3, 1.0, 0.76, and 0.46 wt% by X-ray fluorescence (XRF) spectra. An overloaded sample with a Mo 3.3 wt% loading (denoted as 3.3% $Mo/Fe_2O_3$, see Supplementary Discussion for more details) was also synthesized with the same procedure for comparison.

**Single-atom $W_1/Fe_2O_3$ and $Fe_1/WO_3$ catalysts**. $W_1/Fe_2O_3$ and $Fe_1/WO_3$ were prepared by the same procedure of that of $Mo_1/Fe_2O_3$ except that the $(NH_4)_6H_2W_{12}O_{40}$·$xH_2O$ (MW: 2956.30) and $Fe(NO_3)_3$·$9H_2O$ precursors, and a $\gamma$-$WO_3$ nanoplate support were used, and $Fe_1/WO_3$ was calcined at 400 °C in air for 4 h. The W loadings of $W_1/Fe_2O_3$ were set to be 2.0, 1.5, 1.0, and 0.53 wt% with respect to $\alpha$-$Fe_2O_3$. The Fe loadings of $Fe_1/WO_3$ were set to be 0.28, 0.21, 0.17, 0.07 wt% with respect $\gamma$-$WO_3$. Before the preparation of $Fe_1/WO_3$, the $\gamma$-$WO_3$ nanoplate support was prepared according to the previous report[28]. Briefly, $Na_2WO_4$·$2H_2O$ (1.65 g, 5 mmol) was dissolved in 30 mL de-ionized water, to which an aqueous HCl solution (5 mL, 36–38 wt%) was added under magnetic stirring at room temperature. A $H_2C_2O_4$ (0.45 g, 5 mmol) was introduced into the solution under stirring for 1 h. The resulting dark-yellow precursor solution was transferred into a 50 mL Teflon-lined stainless autoclave, sealed and heated in the oven at 120 °C for 12 h. After cooling to room temperature, the precipitate was collected via centrifugation and further washed with de-ionized water and ethanol, and dried in air at 80 °C. Finally, the obtained powder was calcined at 400 °C for 4 h. Unless mentioned otherwise, the following $Mo_1/Fe_2O_3$, $W_1/Fe_2O_3$, and $Fe_1/WO_3$ refer to 1.3 wt% $Mo/Fe_2O_3$, 2.0 wt% $W/Fe_2O_3$, and 0.28 wt% $Fe/WO_3$, respectively.

**Transmission electron microscopy (TEM) images**. TEM and high-resolution TEM (HRTEM) images were carried out with a JEOL JEM-2100F field-emission gun transmission electron microscope operating at an accelerating voltage of 200 kV and equipped with an ultra-high-resolution pole-piece that provides a point-resolution better than 0.19 nm. Fine powders of the materials were dispersed in ethanol, sonified, and sprayed on a carbon coated copper grid, and then allowed to air-dry.

Aberration-corrected high-angle annular dark filed scanning transmission electron microscopy (AC-STEM) images and energy dispersive X-ray spectroscopy (EDX) elemental mapping were performed at 200 kV with a JEOL ARM-200F FEG TEM equipped with a probe corrector, a high-angle annular dark-field detector, and EDX detector. Fine powders of the materials were dispersed in ethanol,

sonified, and sprayed on a $Si_3N_4$ grid with a size of 8 nm in thickness, and then allowed to air-dry.

**Synchrotron X-ray diffraction (SXRD) patterns**. The X-ray diffraction data were obtained at beamline BL14B1 of the Shanghai Synchrotron Radiation Facility (SSRF) using X-ray with a wavelength of 0.6884 Å. The sample is loaded into a spinning capillary for measurements. Mythen 1 K Si strip linear detector is used for data acquisition. X-ray diffraction (XRD) patterns of some samples were also collected with a Rigaku Ultima-IV diffractometer (Japan) with Cu Kα radiation ($\lambda = 1.5406$ Å).

**X-ray absorption spectra (XAS)**. XAS covers the X-ray absorption near-edge structure (XANES) spectra and extended X-ray absorption fine structure (EXAFS) spectra, which were measured at Mo $K$-edge and Fe $K$-edge at BL14W of the SSRF with an electron beam energy of 3.5 GeV and a ring current of 200–300 mA. Data were collected with a fixed exit monochromator using two flat Si(311) crystals for the Mo $K$-edge XAS measurements or two flat Si(111) crystals for the Fe $K$-edge XAS measurements. Harmonics were rejected by using a grazing incidence mirror. The XANES spectra were acquired at an energy step of 0.5 eV. The EXAFS spectra were collected in a transmission mode using ion chambers filled with $N_2$. The raw data was analyzed using the IFEFFIT 1.2.11 software package. The soft-X-ray absorption spectra at the Mo $L_3$-edge were measured at 4B7A of the Beijing Synchrotron Radiation Facility with an electron beam energy of 2.21 GeV and a ring current of 300–450 mA.

**X-ray photoelectron spectra (XPS)**. XPS were collected on an ESCALAB 250 multifunctional X-ray photoelectron spectroscopy instrument (Thermo Fisher) using a monochromatic Al-Kα X-ray source ($h\nu = 1486.6$ eV). The spectrometer was equipped with a delay-line detector. Spectra were acquired at normal emission with a passing energy of 40 eV. Spectra were all referenced to the C 1s peak at a binding energy of 284.6 eV for each new scan. Data analysis and processing was undertaken using XPSPeak4.1 with Shirley type background.

**Temperature-programmed reduction by hydrogen (H$_2$–TPR) profiles**. H$_2$–TPR was conducted by using an AutoChem II 2950HP auto-adsorption apparatus. The areas of the reduction peaks have been calibrated by the $H_2$ uptakes of the different amounts of CuO. Prior to the reduction process, ~20 mg samples were treated under a $N_2$ atmosphere with a flow rate of 30 mL min$^{-1}$ at 300 °C for 30 min, and then cooled to room temperature under $N_2$ atmosphere. For each run, the sample was reduced in stream of 10.0 vol% $H_2$/Ar (80 mL min$^{-1}$) at a ramp of 2.5 °C min$^{-1}$.

**Raman spectroscopy**. The static Raman measurements of the molecular structures of catalysts were determined on a XploRA confocal spectrometer (Jobin Yvon, Horiba Gr, France). The Raman scattering was excited by an external-cavity diode (785 nm) and coupled with a 50× Olympus microscope objective (Olympus, 0.50 Numerical Aperture). The power of the laser was equal to 9 mW. A 1200 lines per mm diffraction grating places prior to a multichannel charge-coupled device detector (1024 × 256 pixels) was used to collect spectra in a resolution of 3 cm$^{-1}$ with two accumulations at a 10 s acquisition time.

**In situ diffuse reflectance infrared Fourier-transform (DRIFT) spectra**. In situ DRIFT spectra were conducted by accumulating 64 scans at a 4 cm$^{-1}$ resolution in the kubelka-Munk format from 4000 to 1000 cm$^{-1}$ on an FTIR spectrometer (Nicolet iS 50) equipped with a Harrick Scientific DRIFT cell and a mercury-cadmium-telluride MCT/A detector. Prior to each experiment, the catalysts were pretreated at 300 °C in a flow of $N_2$ (30 mL min$^{-1}$) for 0.5 h to remove physically adsorbed water and then cooled to the target temperature under $N_2$ flow to obtain a background spectra which should be deducted from the spectra of samples. After obtaining the background spectra at different temperatures, the catalysts were exposed to a flow of 500 ppm $NH_3$ at 30 °C for 1 h. The desorption process then went on under a flow of $N_2$ (30 mL min$^{-1}$) and was recorded at the corresponding temperature of background spectrum.

**Catalytic evaluations**. SCR activity measurements were performed in a fixed-bed quartz reactor (inner diameter 4 mm) under atmospheric pressure. The feed gas contained 500 ppm NO, 500 ppm $NH_3$, 3.0 vol% $O_2$, and balanced $N_2$. The total flow rate was 1000 mL min$^{-1}$ and 0.1 g sample (40–60 mesh) was used (0.2 g Fe$_1$/WO$_3$ was used to keep the same space velocity as the other samples). The gas hourly space velocity (GHSV) was calculated to be 800,000 h$^{-1}$. Data were recorded by a temperature-programmed procedure at a ramp of 2.5 °C min$^{-1}$. $H_2O$ and $SO_2$ durability measurements were performed in a fixed-bed quartz reactor (inner diameter 8 mm) under atmospheric pressure at 300 °C. The feed gas contained 500 ppm NO, 500 ppm $NH_3$, 3.0 vol% $O_2$, 200 ppm $SO_2$ (when used), 5.0 vol% $H_2O$ (when used) and balanced $N_2$. The total flow rate was 500 mL min$^{-1}$ and 0.6 g sample (40–60 mesh) was used. The concentration of NO in the outlet was continually monitored by an online chemiluminescence NO−NO$_2$−NO$_x$ analyzer (42i-HL,

Thermo Fisher Scientific, Waltham, MA). $N_2$ selectivity ($S_{N2}$) in the SCR process was measured by a Fourier-transform infrared spectrometer (Thermo Scientific Antaris IGS analyzer), and $S_{N2}$ was calculated by a following formula 1:

$$S_{N_2} = \frac{[NO]_{in} + [NH_3]_{in} - [NO]_{out} - [NH_3]_{out} - [NO_2]_{out} - 2[N_2O]_{out}}{[NO]_{in} + [NH_3]_{in} - [NO]_{out} - [NH_3]_{out}} \quad (1)$$

where $[A]_{in}$ and $[A]_{out}$ represent the concentration of A in inlet gas and outlet gas, respectively.

## Data availability

The additional data are provided in the Supplementary Information. All the data that support the findings of this study are available from the corresponding author upon reasonable request.

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

## Acknowledgements

We sincerely thank Prof. Liwu Zhang for the help in the Raman spectroscopy measurement. This work was financially supported by the National Natural Science Foundation of China (21777030 and 21976037), the National Engineering Laboratory for Flue Gas Pollution Control Technology and Equipment (NEL-KF-201903), and the National Engineering Laboratory for Mobile Source Emission Control Technology (NELMS2018B02). The SXRD and X-ray absorption spectra were carried out at Shanghai and Beijing Synchrotron Radiation Facilities.

## Author contributions

Y.C. and X.T. designed and led the experiments. W.Q. and X.L. prepared the catalysts, conducted the experiments, and analyzed the data. J.C. and Y.D. assisted in catalysts preparation and activity measurements. W.Q., X.T., and Y.C. wrote the manuscript. All authors commented on the manuscript.

## Competing interests

The authors declare no competing interests.
