## [Peer Review File · Nature Communications]

Reviewers' comments:

Reviewer #1 (Remarks to the Author):

This paper reported on single-site catalysts of Mo1/Fe2O3, W1/Fe2O3, and Fe1/WO3 for NH3-SCR. Basically, the manuscript was written well, but there are considerable issues in the present form. I recommend publishing after a major revision. I strongly request the below to improve this manuscript.

1. A decrease in surface defect oxygen (Figure S4).

The authors described that 'Blue shade represents a decrease of the surface defect oxygen after the Mo loading' in support information. However, this explanation is confusing. XPS can reflect the state of oxygen on the catalyst. The defect oxygen will not be observed directly by XPS. If the words of 'a decrease of the surface defect oxygen' means that oxygen occupied the lattice, the intensity should become higher for Mo1/Fe2O3.

2. Figure 1F

The overlap of O in Figure 1F is not meaningful. In addition, the color of Mo and O was similar. In this case, the overlap of Fe and Mo is enough to show the dispersion of Fe species as Figure 1F.

3. Distinguish of single Mo site and Mo1-Fe1 site

The distinguish between single Mo site and Mo1-Fe1 site is unclear. It is because we can see the same Mo1-Fe1 structure in single Mo site with the neighbor Fe site in Figure 1H. What is the difference and how to distinguish them?

4. The state of Mo

The XAFS analysis indicated that CN of Mo-O bond was 6. And Mo=O band was confirmed by the Raman. Also, MoO6H was observed by FT-IR on page 8. The MoO6 motif terminated with Mo=O bond will be Mo6+. And MoO6 motif should be octahedra structure in the crystal. But Fe2O3 array is different from octahedra motif. How Mo was anchored in Fe2O3 array as MoO6 motif?

5. Bronsted acid site

I agree that the MoO6H function as Bronsted acid site. However, the importance of Lewis acid site is also pointed out from many publications. How did the author exclude the contribution of Lewis acid site?

6. N balance in catalytic test

The authors monitored NO, NO2 by NOx meter. But the authors didn't check N2, N2O, and NH3 for the catalytic test. The consumption ratio of NH3 and NO should be checked. And the formation of N2O or N2 should be checked to make sure the N balance. Sometimes, the formation of N2O will be observed when we use the new catalysts.

7. The comparison of the reference catalyst

The authors described that the 'calculated TOFs (converted NO molecules per ACS per second) to be $\sim 1.7 \times 10^{-3} \text{ s}^{-1}$ at 270 oC (Figure S16), comparable to the TOF values ($1.3 \times 10^{-3} \text{ s}^{-1}$ at 277 oC, $2.4 \times 10^{-3} \text{ s}^{-1}$ at 323 oC) of V2O5/TiO2' in page 11. The loading amount of V2O5 for the commercial catalyst of V2O5/TiO2 was 0.5 wt% to 9 wt% depending on the application. And the TOF of Mo1/Fe2O3 was comparable to V2O5/TiO2. In the case of ca 1wt% V2O5/TiO2, it shows the almost full conversion at more than 350 oC. Why the catalytic activity of the Mo1/Fe2O3 catalyst was lower than that of the V2O5/TiO2 although the TOF value was similar.

8. The comparison of TOF

The authors reported Mo₁/Fe₂O₃, W₁/Fe₂O₃, and Fe₁/WO₃. The simple comparison of the TOF value of these catalysts should be added for the benefit of readers.

9. Water and SO₂ tolerance

The influence of water on catalytic activity should be tested. The addition of water affects both catalytic activity and reaction mechanism.

Also the SO₂ tolerance of this material should be tested and mentioned.

Reviewer #2 (Remarks to the Author):

This is a very nice paper describing the synthesis, characterization and reactivity of atomically dispersed catalysts for NO SCR. Specific focus is given to the need of active sites with a combination of acidity and redox properties. To achieve this, acidic species (MoO_x or WO_x) are dispersed on a redox active support (Fe₂O₃), or a redox active site (FeO_x) was anchored on an acidic support WO₃. The characterization is very nice and the linear site density-reactivity relationships are convincing in assigning the active sites and their homogeneity. The paper is recommended for publication with only minor issues to address.

1. The paper is beautifully written. Although, some wording could be improved. For example, between pg 3 and 4 the use of "plenty of" should be modified.

2. The STEM data in Figure 4D is not as convincingly assigned as was seen for Figure 1g. Specifically, it seems like many more W species may exist in neighboring sites just based on scattering intensity. This should be clarified with some discussion.

3. The wording of dinuclear should be clarified. For example, in M on Fe₂O₃ the M atom has 3 neighboring Fe species, so the idea of a dinuclear species doesn't completely make sense. For example, recent work has specifically synthesized dinuclear active sites that don't involve cations from the support (see ACS Catal. 2019, 9, 12, 10899-10912). This comment is not meant to take away from the current paper, just to suggest that the authors clarify the structure of the sites described here compared to other recent reports.

I. Reply to the reviewers

We thank the editor and the reviewers for carefully reviewing our manuscript. We have revised the manuscript carefully, according to the reviewers' comments. Below is a point-by-point response to the reviewers' comments.

Reviewer #1

This paper reported on single-site catalysts of $\text{Mo}_1/\text{Fe}_2\text{O}_3$, $\text{W}_1/\text{Fe}_2\text{O}_3$, and Fe_1/WO_3 for NH_3 -SCR. Basically, the manuscript was written well, but there are considerable issues in the present form. I recommend publishing after a major revision. I strongly request the below to improve this manuscript.

Comment 1.1: A decrease in surface defect oxygen (Figure S4).

The authors described that 'Blue shade represents a decrease of the surface defect oxygen after the Mo loading' in Supporting Information. However, this explanation is confusing. XPS can reflect the state of oxygen on the catalyst. The defect oxygen will not be observed directly by XPS. If the words of 'a decrease of the surface defect oxygen' means that oxygen occupied the lattice, the intensity should become higher for $\text{Mo}_1/\text{Fe}_2\text{O}_3$.

Reply 1.1: Thank the reviewer for the good comments. We agreed that the explanation about the O 1s XPS is confusing, which possibly originates from the different definitions of defect oxygen. Some researchers defined the oxygen vacancy as defect oxygen (*ACS Appl. Mater. Interfaces*, **2014**, 6, 12505–12514; *ACS Appl. Mater. Interfaces*, **2016**, 8, 33765-33774), while others defined surface oxygen with the low coordination number as the defect oxygen (*Phys. Chem. Chem. Phys.*, **2000**, 2, 1319-1324; *Electrochim. Acta*, **2017**, 229, 229-238; *ACS Sustainable Chem. Eng.*, **2019**, 7, 12117–12124). We agreed with the reviewer that it is more suitable to define oxygen vacancy as the defect oxygen, which thus cannot be observed directly by XPS. As a result, we modified the manuscript as “As further analyzed by the selected-area intensity surface plot and the corresponding structural model (Figures 1H and S3)”

and deleted the Figure S4 in the revised Supporting Information.

Comment 1.2: Figure 1F.

The overlap of O in Figure 1F is not meaningful. In addition, the color of Mo and O was similar. In this case, the overlap of Fe and Mo is enough to show the dispersion of Fe species as Figure 1F.

Reply 1.2: According to the reviewer's suggestion, we have deleted the overlap of O in Figure 1F, and thus the dispersions of Mo and Fe species are clearly displayed in the revised Figure 1F.

Comment 1.3: Distinguish of single Mo site and Mo₁-Fe₁ site.

The distinguish between single Mo site and Mo₁-Fe₁ site is unclear. It is because we can see the same Mo₁-Fe₁ structure in single Mo site with the neighbor Fe site in Figure 1H. What is the difference and how to distinguish them?

Reply 1.3: To clearly distinguish the single Mo site from the Mo₁-Fe₁ site, we have changed the 'single Mo site' to the single Mo atom, according to the reviewer's comments in the revised manuscript. Meanwhile, we have also changed 'single-site' to 'single-atom' in the revised manuscript. As a consequence, the 'dinuclear Mo₁-Fe₁ site' is used as the catalytically active site in the SCR process, which is consistent with the recent research (*Angew. Chem. Inter. Ed.*, **2019**, 58, 12609-12616).

Comment 1.4: The state of Mo.

The XAFS analysis indicated that CN of Mo-O bond was 6. And Mo=O band was confirmed by the Raman. Also, MoO₆H was observed by FT-IR on page 8. The MoO₆ motif terminated with Mo=O bond will be Mo⁶⁺. And MoO₆ motif should be octahedra structure in the crystal. But Fe₂O₃ array is different from octahedra motif. How Mo was anchored in Fe₂O₃ array as MoO₆ motif?

Reply 1.4: As commented by the reviewer, the XAFS analysis indicated that CN of Mo-O bond was 6, and Mo=O band was confirmed by the Raman. Also, MoO₆H was

observed by FT-IR on page 8. We determined the oxidation state of Mo species to be Mo^{5+} in $\text{Mo}_1/\text{Fe}_2\text{O}_3$ by combining the Mo 3d X-ray photoelectron spectra (Figure S7) with the Mo L_3 -edge X-ray absorption spectra (Figure 2B). Although the oxidation states of Mo species are associated with the Mo=O bonding model, the oxidation states of Mo in MoO_6 motif with Mo=O bonds might be +5 or +6 (*Appl. Surf. Sci.*, **1989**, 40, 179-181; *Appl. Catal. B*, **1998**, 3, 245-258; *Science*, **2015**, 348, 686-690).

Generally, MoO_6 motif is the octahedral structure in the α - MoO_3 crystal, as confirmed by the Mo L_3 -edge X-ray absorption spectra of α - MoO_3 in Figure 2B. Meanwhile, the FeO_6 motif in the α - Fe_2O_3 crystal is also the octahedral structure (*Chem. Mater.*, **2011**, 23, 14, 3255-3272), although FeO_6 octahedral motif is slightly different from that of MoO_6 octahedral motif due to the different size of metal ions and their electronic configurations. As for $\text{Mo}_1/\text{Fe}_2\text{O}_3$, we compared the Mo L_3 -edge X-ray absorption spectra of $\text{Mo}_1/\text{Fe}_2\text{O}_3$ with two reference samples with different symmetry, i.e. α - MoO_3 with a MoO_6 octahedral symmetry and $\text{Fe}_2(\text{MoO}_4)_3$ with a MoO_4 tetrahedral symmetry, in Figure 2B. The results demonstrated that the MoO_6 motif of $\text{Mo}_1/\text{Fe}_2\text{O}_3$ is a distorted octahedral structure, as confirmed by these data of the AC-STEM image (Figure 1), the EXAFS data (Figure 2A) and the Raman evidence (Figure S6).

On the α - $\text{Fe}_2\text{O}_3(001)$ surface (Figure S2D), there are abundant three-fold hollow sites formed by three surface lattice oxygen atoms, which serve as suitable sites for anchoring Mo ions. As confirmed in the manuscript, the anchored Mo ion has three surface dangling bonds besides three Mo-O bonds formed with three oxygen ions of the anchoring site to form a distorted MoO_6 octahedral structure. Thus, Mo can be anchored as the distorted MoO_6 octahedral structure motif on the α - $\text{Fe}_2\text{O}_3(001)$ surface, though we did not know the accurate geometric structure.

Comment 1.5: Brønsted acid site.

I agree that the MoO_6H functions as Brønsted acid site. However, the importance of Lewis acid site is also pointed out from many publications. How did the author

exclude the contribution of Lewis acid site?

Reply 1.5: Brønsted acid site plays an important role in the SCR process, which has been evidenced by many researchers (*Science*, **1994**, 265, 1217-1219; *J. Catal.*, **1995**, 151, 226-240; *J. Catal.*, **2017**, 346, 188-197; *J. Am. Chem. Soc.*, **2017**, 139, 15624-15627). According to the reaction mechanism (*J. Phys. Chem.*, **1987**, 91, 5921-5927), Brønsted acid sites and Lewis acid sites could transform from one to the other during the SCR reaction process (*J. Catal.*, **2020**, 382, 269–279), indicating that both Brønsted acid sites and Lewis acid sites participate in the SCR reactions. According to the reviewer’s suggestion, we have added the *in situ* DRIFT spectra collected at the SCR reaction temperature of 250 °C into the revised Supporting Information. As shown in Figure S10, the Lewis acid sites can be observed during the SCR reaction process. Accordingly, the manuscript has been revised as “Thus, each isolated Mo ion due to the formation of a MoO₆H species can provide one Brønsted acid site²³, which can transform to the Lewis acidic site during the SCR reactions²⁴ (Figure S10).”

Figure R1. DRIFT spectra of NH₃ adsorption on Mo₁/Fe₂O₃ and α-Fe₂O₃ at 250 °C.

Comment 1.6: *N balance in catalytic test.*

The authors monitored NO, NO₂ by NO_x meter. But the authors didn't check N₂, N₂O, and NH₃ for the catalytic test. The consumption ratio of NH₃ and NO should be checked. And the formation of N₂O or N₂ should be checked to make sure the N balance. Sometimes, the formation of N₂O will be observed when we use the new catalysts.

Reply 1.6: As the reviewer suggested, we have recorded the NH₃ and NO conversions of Mo₁/Fe₂O₃ together with α-Fe₂O₃ in the SCR process. As shown in Figure R2A, the conversions of NH₃ and NO over Mo₁/Fe₂O₃ are almost the same as each other, in the whole temperature range. We further recorded the concentrations of N₂O (Figure R2B) and the concentrations of NO₂ can be ignored, from which we calculated the N₂ selectivity as shown in Figure R2C. The results demonstrated that Mo₁/Fe₂O₃ has high catalytic activity and selectivity. In contrast, for α-Fe₂O₃, the conversion of NO decreased rapidly and became lower than that of NH₃ above 300 °C, as shown in Figure R2D, which suggested that the NH₃ was oxidized to NO by α-Fe₂O₃ at high temperatures. Meanwhile, as the temperature increases, N₂O concentration over α-Fe₂O₃ increases (Figure R2B), leading to the low selectivity to N₂ (Figure R2C). We have added these results into the revised Supporting Information as Figure S12.

Figure R2. SCR performance as a function of temperature (*T*): (A) NO and NH₃

conversion over Mo₁/Fe₂O₃. (B) N₂O concentration over α-Fe₂O₃ and Mo₁/Fe₂O₃. (C) N₂ selectivity over α-Fe₂O₃ and Mo₁/Fe₂O₃. (D) NO and NH₃ conversion over α-Fe₂O₃. Reaction conditions: 500 ppm NO, 500 ppm NH₃, 3 vol% O₂, balance N₂, and GHSV 66,000 h⁻¹.

Comment 1.7: The comparison of the reference catalyst.

The authors described that the ‘calculated TOFs (converted NO molecules per ACS per second) to be $\sim 1.7 \times 10^{-3} \text{ s}^{-1}$ at 270 °C (Figure S16), comparable to the TOF values ($1.3 \times 10^{-3} \text{ s}^{-1}$ at 277 °C, $2.4 \times 10^{-3} \text{ s}^{-1}$ at 323 °C) of V₂O₅/TiO₂’ in page 11. The loading amount of V₂O₅ for the commercial catalyst of V₂O₅/TiO₂ was 0.5 wt% to 9 wt% depending on the application. And the TOF of Mo₁/Fe₂O₃ was comparable to V₂O₅/TiO₂. In the case of ca 1wt% V₂O₅/TiO₂, it shows the almost full conversion at more than 350 °C. Why the catalytic activity of the Mo₁/Fe₂O₃ catalyst was lower than that of the V₂O₅/TiO₂ although the TOF value was similar.

Reply 1.7: Depending on the reaction conditions such as the reaction temperatures and the gas hourly space velocity (GHSV), catalytic activities are often different over different catalyst systems, although they hold the same TOF. In particular, one important factor governing the different activities between Mo₁/Fe₂O₃ and V₂O₅/TiO₂ is the difference of their GHSVs. In current work, GHSV was calculated to be as high as 600,000 mL g⁻¹ h⁻¹, whereas the reported values of the V₂O₅/TiO₂ catalysts are only 22,500 mL g⁻¹ h⁻¹ (*J. Catal.*, **1994**, 147, 241–249) and 45,000 mL g⁻¹ h⁻¹ (*J. Catal.*, **1999**, 187, 419–435). Therefore, it is the main reason why the catalytic activity of the Mo₁/Fe₂O₃ catalyst was lower than that of the V₂O₅/TiO₂ although the TOF value was similar.

Comment 1.8: The comparison of TOF.

The authors reported Mo₁/Fe₂O₃, W₁/Fe₂O₃, and Fe₁/WO₃. The simple comparison of the TOF value of these catalysts should be added for the benefit of readers.

Reply 1.8: As the reviewer suggested, the TOF values of Mo₁/Fe₂O₃, W₁/Fe₂O₃, and

Fe₁/WO₃ were calculated from the 1.3 wt% Mo/Fe₂O₃, 2.0 wt% W/Fe₂O₃, and 0.28 wt% Fe/WO₃, respectively (Figure R3), which were also added in the revised Supporting Information as Figure S24.

Figure R3. TOFs over Mo₁/Fe₂O₃, W₁/Fe₂O₃, and Fe₁/WO₃ at 270 °C. Reaction conditions: 500 ppm NO, 500 ppm NH₃, 3 vol% O₂, balance N₂, and GHSV 800,000 h⁻¹.

Comment 1.9: Water and SO₂ tolerance.

The influence of water on catalytic activity should be tested. The addition of water affects both catalytic activity and reaction mechanism. Also the SO₂ tolerance of this material should be tested and mentioned.

Reply 1.9: According to the reviewer comments, we have tested the effect of water on catalytic activity of Mo₁/Fe₂O₃ after reaching a steady state. When H₂O was added into the feed gas, the catalytic activity rapidly decreased from 92% to 70%, and then remained stable. However, after cutting off the supply of water, the catalytic activity rapidly restored to the original level. This result implied that the inhibiting effect of H₂O on the activity of Mo₁/Fe₂O₃ was reversible. We have also tested the effect of SO₂. Nearly no decrease of activity was observed after the addition of SO₂, which indicated that Mo₁/Fe₂O₃ had a strong SO₂ tolerance. In real flue gases from industry, water and SO₂ usually co-exist, so we have tested the effect of H₂O and SO₂. The

result indicated the excellent H₂O and/or SO₂ durability of Mo₁/Fe₂O₃. We have added these results into the revised Supporting Information as Figure S13, and we revised the manuscript as: “and the Mo₁/Fe₂O₃ also showed excellent H₂O and/or SO₂ durability (Figure S13)”.

Figure R4. Effect of H₂O and SO₂ on catalytic activity over Mo₁/Fe₂O₃ at 300 °C. Reaction conditions: 500 ppm NO, 500 ppm NH₃, 3 vol% O₂, 200 ppm SO₂ (when used), 5 vol% H₂O (when used), balance N₂, and GHSV 66,000 h⁻¹.

Reviewer #2

This is a very nice paper describing the synthesis, characterization and reactivity of atomically dispersed catalysts for NO SCR. Specific focus is given to the need of active sites with a combination of acidity and redox properties. To achieve this, acidic species (MoO_x or WO_x) are dispersed on a redox active support (Fe₂O₃), or a redox active site (FeO_x) was anchored on an acidic support WO₃. The characterization is very nice and the linear site density-reactivity relationships are convincing in assigning the active sites and their homogeneity. The paper is recommended for publication with only minor issues to address.

Comment 2.1: *The paper is beautifully written. Although, some wording could be*

improved. For example, between pg 3 and 4 the use of “plenty of” should be modified.

Reply 2.1: Thank the reviewer for the good comment. We have replaced “plenty of” by “abundant” in the revised manuscript.

Comment 2.2: The STEM data in Figure 4D is not as convincingly assigned as was seen for Figure 1g. Specifically, it seems like many more W species may exist in neighboring sites just based on scattering intensity. This should be clarified with some discussion.

Reply 2.2: As the reviewer suggested, we have taken the image intensity line scans in the areas, where W species seem to exist in neighboring sites. The intensity profiles revealed that the most W exists as isolated single atoms (Figure R5). We have added the result into the revised Supporting Information as Figure S19. It might also be the existence of adjacent W species, if so, these adjacent W species should not make a great contribution to our conclusion owing to a low activity in SCR reactions, as testified by the very low conversion on the two adjacent W sites of WO_3 in our work (Figure S26).

Figure R5. (A) AC-STEM image of $\text{W}_1/\text{Fe}_2\text{O}_3$. (B) The image intensity line scans along the directions a and b shown in A.

Comment 2.3: *The wording of dinuclear should be clarified. For example, in M on Fe₂O₃ the M atom has 3 neighboring Fe species, so the idea of a dinuclear species doesn't completely make sense. For example, recent work has specifically synthesized dinuclear active sites that don't involve cations from the support (see ACS Catal. 2019, 9, 12, 10899-10912). This comment is not meant to take away from the current paper, just to suggest that the authors clarify the structure of the sites described here compared to other recent reports.*

Reply 2.3: As the reviewer mentioned, on the Mo₁/Fe₂O₃ or W₁/Fe₂O₃ surfaces, each isolated Mo or W ion connects to three neighboring Fe ions bridged by the lattice oxygen (Figure S2), and thus one active catalytic site (ACS) might contain more than one Fe ions besides one Mo or W ion. To rule out this possibility, we anchored single Fe ions on (001) surfaces of γ -WO₃ nanosheets with a square morphology to achieve a single-atom Fe₁/WO₃ catalyst (Figure S25). As the number of the single Fe ions increases, X_{NO} over Fe₁/WO₃ linearly increases, the trend of which is the same as that of W₁/Fe₂O₃ (Figure 4E). As a consequence, only single Fe ion is enough for one ACS of Mo₁/Fe₂O₃ or W₁/Fe₂O₃. According to the reviewer's suggestion, we have also cited the paper (ACS Catal., 2019, 9, 12, 10899-10912) as Reference 17 to clarify the concept of dinuclear metal sites, as described in the revised manuscript: "An ideal structure is to fabricate dinuclear metal sites on supports¹⁷, which allows it to function as dual sites catalyzing SCR reaction, but it is a formidable task to synthesize such a catalyst."

REVIEWERS' COMMENTS:

Reviewer #1 (Remarks to the Author):

Now my questions were became clear.

Reviewer #2 (Remarks to the Author):

The authors nicely addressed all concerns. The paper can be published.

Response to the reviewers

We thank the reviewers for carefully reviewing our manuscript. Below is a point-by-point response to the reviewers' comments.

Reviewer #1

Now my questions were became clear.

Response 1: Thank you very much for your previous valuable comments, positive evaluation and publication recommendation on this work.

Reviewer #2

The authors nicely addressed all concerns. The paper can be published.

Response 2: Thank you very much for your positive evaluation, all the previous valuable comments and publication recommendation on this work.